# AMP activated kinase negatively regulates hepatic Fetuin-A via p38 MAPK-C/EBPβ/E3 Ubiquitin Ligase Signaling pathway

Vishal Kothari[1], Jeganathan Ramesh Babu[1], Suresh T. Mathews[1,2]*

1 Department of Nutrition and Dietetics, Boshell Diabetes and Metabolic Diseases Research Program, Auburn University, Auburn, AL, United States of America, 2 Department of Nutrition and Dietetics, Samford University, Birmingham, AL, United States of America

* smathew1@samford.edu

**Data Availability Statement:** All relevant data are within the manuscript and its Supporting Information files

**Funding:** This study was supported by the United States Department of Agriculture, National Institute

## Abstract

Fetuin-A (Fet-A) is a liver-secreted phosphorylated protein, known to impair insulin signaling, which has been shown to be associated with obesity, insulin resistance, and incident diabetes. Fet-A interacts with the insulin-stimulated insulin receptor (IR) and inhibits IR tyrosine kinase activity and glucose uptake. It has been shown that high glucose increases Fet-A expression through the ERK1/2 signaling pathway. However, factors that downregulate Fet-A expression and their potential mechanisms are unclear. We examined the effect of AMP-activated protein kinase (AMPK) on high-glucose induced Fet-A expression in HepG2 cells, Hep3B cells and primary rat hepatocytes. High glucose increased Fet-A and phosphorylated (Ser312) fetuin-A (pFet-A) expression, which are known to impair insulin signaling. AICAR-induced AMPK activation significantly down-regulated high glucose-induced Fet-A expression and secretion of pFet-A while treatment with Compound C (AMPK inhibitor), SB202190 (p38 MAPK inhibitor) or p38 MAPK siRNA transfection prevented AICAR-induced downregulation of Fet-A expression. In addition, activation of p38 MAPK, by anisomycin, decreased the hepatic expression of Fet-A. Further, we our studies have shown that short-term effect of AICAR-treatment on Fet-A expression was mediated by proteosomal degradation, and long-term treatment of AICAR was associated with decrease in hepatic expression of C/EBP beta, an important transcription factor involved in the regulation of Fet-A. Taken together, our studies implicate a critical role for AMPK-p38 MAPK-C/EBPb-ubiquitin-proteosomal axis in the regulation of the expression of hepatic Fet-A.

## Introduction

The liver is known to play an important role in the ongoing epidemic of type 2 diabetes, nonalcoholic fatty liver disease (NAFLD), and cardiovascular disease. In these conditions, the liver is involved in increased glucose production and dysregulated lipoprotein metabolism 2 [1,2]. Analogous to adipokines and myokines, recent studies have suggested that the liver may regulate glucose homeostasis by modulating the sensitivity/resistance of peripheral tissues, by way

of Food and Agriculture grant ALA043-1-08033 (to
S.T.M) and Malone-Zallen Graduate Research
Fellowship to Vishal Kothari. There was no
additional external funding received for this study.
The funders had no role in study design, data
collection and analysis, decision to publish, or
preparation of the manuscript.

**Competing interests:** The authors have declared
that no competing interests exist.

**Abbreviations:** Fet-A, fetuin-A; pFet-A,
phosphorylated (Ser312) fetuin-A; AMPK, AMP-
activated kinase.

of the production of secretory proteins, termed hepatokines [3–5]. Fetuin-A (Fet-A, also known as alpha-2-HS-glycoprotein in humans) is a major hepatokine, originally identified in 1944; but its importance in physiology has only recently been appreciated [6]. Fet-A functions as an important component of diverse normal and pathological processes, including insulin resistance, vascular calcification and bone metabolism [7].

Fet-A shares amino-acid sequence similarity to insulin receptor tyrosine kinase [8] and type-II transforming growth factor-ß (TGF-ß) receptor [9], and thus been proposed as a natural inhibitor of the insulin-signaling pathway and an antagonist of TGF-ß. Recently, we have shown that phosphorylation status of fetuin-A is critical for inhibition of insulin action, correlating with obesity and insulin resistance [10]. Previously, we also demonstrated that Fet-A interacts with the activated insulin receptor and inhibits IR and IRS-1 phosphorylation in liver and skeletal muscle. Fet-A showed relative specificity for IR inhibition, without affecting epidermal growth factor or insulin-like growth factor1-induced cognate receptor autophosphorylation and TK activity [11,12]. Fet-A impairs insulin-mediated glucose uptake in C2C12 and L6-GLUT4myc myotubes by down-regulating GLUT-4 translocation to the plasma membrane through decreased phosphorylation of Akt and AS160 [10,13].

Serum Fet-A concentrations were shown to be up-regulated in high-fat diet induced obese animals [14] and in patients with NAFLD [15]. On the contrary, exercise and dietary lifestyle interventions resulted in decline in serum Fet-A and phosphorylated (Ser312) fetuin-A levels commensurate with improvement in NAFLD and a decline in body weight [16]. Fet-A null mice show improved insulin signaling and prevented weight gain when fed a high-fat diet, indicating the physiological relevance of *in vitro* observations [17]. Epidemiological studies have shown that elevated levels of serum Fet-A are associated with obesity, type 2 diabetes mellitus, and metabolic syndrome [15,16,18–20]. Recent studies have implicated Fet-A in adipocyte dysfunction [14], toll-like receptor 4 activation [21], migration and polarization of macrophage [22], and hepatocyte triacylglycerol accumulation [15]. However, regulation of Fet-A and its phosphorylated forms in relation to insulin resistance is not fully understood.

Fet-A gene expression has been shown to be downregulated by tumor necrosis factor-α [23] and by other pro-inflammatory cytokines, including IL-6 and IL-1β, in rat and human hepatocytes [24]. In contrast, high glucose [25] and free fatty acids [14] upregulate hepatic Fet-A gene expression via ERK1/2 and NFkB pathways, respectively. Furthermore, glucocorticoids [26], endoplasmic reticulum [ER] stress [27] and estrogen [28] have been shown to increase Fet-A expression. However, factors that downregulate Fet-A expression and their potential mechanisms are unclear. Recent studies examining the effect of AMPK activation on Fet-A synthesis and secretion are somewhat equivocal. Haukeland *et al* demonstrated that metformin treatment significantly decreased plasma Fet-A compared to placebo in NAFLD patients and dose-dependently decreased Fet-A secretion in HepG2 cells [15]. Further, Jung *et al* demonstrated that salsalate reduced Fet-A expression through the AMPK-NFkB pathway and improved palmitate-induced steatosis and impairment of lipid metabolism [29]. On the other hand, Mori *et al* found that pioglitazone, but not metformin treatment for 6 months, decreased serum Fet-A, in type 2 diabetic patients [30]. Further, these authors reported that, unlike metformin, pioglitazone acts directly to downregulate Fet-A in Fao rat hepatoma cells [31].

To clarify the role of AMPK activation on Fet-A expression, we examined the effect of AICAR and metformin on high glucose-induced Fet-A and pFet-A expression in HepG2 cells, Hep3B cells and primary rat hepatocytes. We also investigated the pathway by which AMPK activation can regulate Fet-A expression. Here, we demonstrate that AICAR regulates high glucose mediated hepatic Fet-A expression through AMPK/p38MAPK pathways. Further, we demonstrate that the effect of AICAR is mediated through a combination of increased degradation of Fet-A and a decrease in the synthesis of Fet-A mediated through C/EBP beta.

## Materials and methods

### Reagents and antibodies

All cell culture materials were obtained from VWR International (Radnor, PA) or Life Technologies (Grand Island, NY). AICAR were purchased from Cayman Chemical Company (Cat # 10010241, Ann Arbor, MI). Recombinant Fet-A protein and human Fet-A ELISA were procured from BioVendor (Cat # RD191037100, Asheville, NC). Metformin (Cat # ALX-270-432-G005), Compound C (Cat # BML-EI369-0005; AMPK inhibitor), SB202190 (Cat # BML-EI294-0001; p38 MAPK inhibitor), MG-132 (Cat # BML-PI102-0005; proteasome inhibitor) were obtained from Enzo Life Science. Anisomycin (Cat # 1290) purchase from Tocris Bioscience. (Anti-pAMPK (Thr172, Cat # 2531), anti-AMPK (Cat # 2532), anti-pERK1/2 (The202/Tyr204, Cat # 9101), anti-pJNK (Cat # 9251), anti-pAKT (Ser473, Cat # 9271)) and anti-pGSK3 (Ser21, Cat # 9331) antibodies were purchased from Cell Signaling (Danvers, MA). Antibodies against C/EBPβ (C-19, Cat # sc-150), p38 MAPK (Cat # sc-81621); pP38 MAPK (Thr 180/Tyr182, Cat # sc-166182), ubiquitin (Cat # sc-8017) and GAPDH (Cat # sc-47724) were purchased from Santa Cruz Biotechnology (Santa Cruz, CA). Anti-Fet-A antibody was from R&D Systems (Cat # AF1563; Minneapolis, MN). Phosphorylated (Ser312) Fet-A was detected using a custom-generated antibody that specifically recognized phosphorylation on Ser312-Fet-A [32]. p38 MAPK siRNA (short interfering RNA; Entrez-Gene ID#1432) was purchased from OriGene Technologies (Cat # SR412160; Rockville, MD). All other chemicals were purchased from Sigma-Aldrich (St. Louis, MO).

### Cell lines and primary culture

HepG2 and Hep3B human hepatocyte-derived cell lines were purchased from American Type Culture Collection (Manassas, VA). HepG2 cells were cultured in DMEM (Dulbecco's Modified Eagle's Medium; Cat# 12320032) containing 10% (v/v) FBS (fetal bovine serum), penicillin, streptomycin and neomycin (Cat# P9032) in a humidified 5% CO2 atmosphere at 37°C. Hep3B cells were cultured in MEM (minimal essential medium, Cat # A1048901) containing non-essential amino acids supplemented with 2 mM L-glutamine (Cat# G8540), 1 mM sodium pyruvate (Cat# P5280), 10% (v/v) FBS (Cat# 12103C) and antibiotics. Fresh primary rat hepatocytes, plated as a monolayer in 6-well plates, were procured from Triangle Research Labs (Research Triangle Park, NC) and maintained in hepatocytes maintenance media (Cat # MM250).

### Cell culture treatment

Confluent HepG2 or Hep3B cells were subcultured by trypsinization and subsequently seeded in 6-well culture plates. Unless otherwise indicated, cells were serum-starved overnight and incubated with low glucose (5.5mM) or high glucose (25.5 mM) for 12 h. Concomitantly, cells were treated with AICAR (0, 0.5, 1, 2 mM), metformin (2 mM), or anisomycin (0.5 μg/ml). When inhibitors were used, cells were pre-incubated with Compound C (10 μM), SB202190 (25 μM) or MG-132 (0.5 μM) for 1 h before treatment with high glucose and AICAR. Primary rat hepatocytes were treated with either AICAR or anisomycin at the indicated dose for 12 h. At the end of the incubation period, medium was collected, or cells were lysed and centrifuged at 10,000 x g for 20 min.

### Immunoprecipitation and Western blot analysis

Following each treatment, the cells were washed twice with ice-cold phosphate buffer saline (PBS) buffer and lysed in the buffer supplemented with 50 mM HEPES (Cat# H3375), 1%

Triton X 100 (Cat# X100), 10 mM EDTA (Cat# E6758), 100 mM sodium pyrophosphate (Cat# 221368), 100 mM sodium fluoride, 10 mM sodium orthovanadate, and protease inhibitor cocktail (Cat# M221; Amresco, Solon, OH). The cells were lysed with SDS lysis buffer to detect covalent interaction of ubiquitin and Fet-A (Triton lysis buffer containing 2% SDS). The protein content in total cell lysates was determined using the Pierce 660 Protein Assay kit (Cat# 22662; Life Technologies) except those samples containing SDS, which were estimated by detergent-compatible assay (Cat # 5000111; Bio-Rad). Cell lysates or culture supernatant medium were mixed with a sample loading buffer and separated on 8% or 4–20% SDS-PAGE gel (Cat # NG10420; NuSep Inc, GA). For immunoprecipitation, cell lysates (500 μg) were diluted in lysis buffer and incubated with 4 μg of primary antibody. The immunoprecipitates were collected with protein A-Sepharose beads (Cat # GE17-0780-01; Sigma) overnight at 4°C and then washed three times with PBS. Samples were boiled in SDS-PAGE sample buffer and resolved on either 4–20% SDS-PAGE. Proteins were transferred to nitrocellulose membranes and incubated with appropriate antibodies. Protein bands were visualized by UVP BioImaging and VisionWorks software package (UVP, Upland, CA) using SuperSignal West Dura Extended Duration substrate (Cat # 34075; Pierce, Rockford, IL) and SuperSignal West Femto maximum sensitivity substrate (Cat # 34095; Pierce, Rockford, IL). Relative area densities were quantified using the UN-SCAN IT software package, v.6.1 (Silk Scientific, Orem, UT).

## ELISA

The total amount of Fet-A secreted into the culture media was determined using a Human Fet-A (BioVendor R&D, Asheville, NC) ELISA kit following the manufacturer´s instructions.

## Real-time PCR

Total RNA was isolated from HepG2 cells using the RNeasy Mini Kit (Qiagen, Valencia, CA), reverse transcribed to cDNA using iScript cDNA synthesis kit (Bio-Rad), and real-time gene expression analysis was performed using iQ SYBR green supermix (Bio-Rad) on a MyiQ Real-time PCR detection system. The following primer sets were used, Fet-A forward primer- 5'-ACG TGG TCC ACA CTG TCA AA-3', Fet-A reverse primer- 5'-CGC AGC TAT CAC AAA CTC CA-3', phosphoenolpyruvate carboxy kniase (PEPCK), forward: GGG TGC TAG ACT GGA TCT GC-3', PEPCK reverse 5'-GAG GGA GAA CAG CTG AGT GG-3'; β-actin forward primer- 5'-CCT CTA TGC CAA CAC AGT GC-3' and β-actin reverse primer- 5'-CAT CGT ACT CCT GCT TGC TG-3'. Reaction conditions were as follows: 95°C, 3:0 min; 95°C, 0:15 min, 60°C, 0:30 min, 72°C, 0:30 min, repeated 40×; 55°C, 0:10 min, repeated 80×. Expression levels were normalized to β-actin and gene expression was calculated as $2^{-\Delta\Delta CT}$ and expressed as fold change. All assays were carried out in triplicate.

## Glucose production assay

HepG2 cells were treated with 0.5 μM dexamethasone and 0.1 mM 8-CTP-cAMP (Dex/cAMP), various concentrations of recombinant Fet-A or 100 nM insulin in glucose production buffer (glucose-free DMEM medium, pH 7.4, containing 20 mM sodium lactate and 2 mM sodium pyruvate, without phenol red) for 5 h. Cells were washed with Dulbecco's PBS, and then incubated for 3 h in glucose production buffer with the same concentrations of Dex/cAMP, insulin and Fet-A. Glucose production was assayed by measuring glucose concentration in the medium as described before [33]. Corrections for cell number were made based on the protein concentration, assayed using Bio-Rad's Bradford protein assay reagent (Bio-Rad, Hercules, CA).

### Transfection

To silence the p38 MAPK gene expression, Mapk14 mouse siRNA oligo duplex (Locus ID 26416) was used (OriGene). HepG2 cells were plated in 12-well culture plate dishes. and transfected with 10nM siRNA with siTran 1.0 (OriGene) transfection reagent according to the manufacturer's instructions. After 36 h of transfection, cells were treated with AICAR (2 mM) for 12 h. At the end of treatment cell were lysed and used for Western blot analysis. p38 MAPK knockdown was confirmed by Western blot analysis.

### Statistical analysis

Results are expressed as Mean ± Standard Error of the Mean (SEM). Comparisons between various treatments and/or groups were carried out using the unpaired Student's t-test or one-way Analysis of Variance (ANOVA) where appropriate. Statistical analyses were performed using GraphPad Prism 6 (GraphPad, San Diego, CA). Differences were considered statistically significant when $p < 0.05$.

## Results

### AMPK activation downregulates glucose-induced Fet-A expression in HepG2 cells

Fet-A is a secretary protein, known to impair adipocyte and skeletal muscle function [13,14]. Our recent observations suggest that phosphorylation status of fetuin-A is critical for inhibition of insulin action [10]. Here, to determine the effect of Fet-A on insulin signaling in HepG2 human hepatoma cells, confluent cells were treated with varying concentrations of recombinant Fet-A. As depicted in Fig 1A–1C, recombinant Fet-A inhibited insulin-induced phosphorylation of AKT and GSK3 in a dose dependent manner. Next, we examined effect of Fet-A on downstream insulin signaling pathway of gluconeogenesis and hepatic glucose production. We observed that Fet-A impaired inhibitory response of insulin on dexamethasone-induced gene expression of *Pepck* (Fig 1D) and glucose production (Fig 1E).

Next, to elucidate changes in Fet-A expression and secretion from the hepatocytes in response to high glucose, HepG2 cells were cultured using low normal glucose (5mM), high-glucose (25mM), or mannitol (25mM) for osmolarity control. HepG2 cells cultured in high glucose expressed higher levels of Fet-A and pFet-A as compared to low glucose and mannitol (Fig 2A). However, high glucose significantly increased the secretion of pFet-A into the media, but not Fet-A (Fig 2A), indicating majority of the Fet-A was secreted in phosphorylated form. Increase in Fet-A expression in HepG2 cells by high-glucose (12hr) was also associated with the impairment of insulin signaling indicated by reduced insulin-induced phosphorylation of GSK3 (S1A Fig), consistent with previous observations [34–36].

Evidence for the role of AMPK activation on Fet-A expression is currently unclear. In addition, the effect of AMPK on high glucose-induced pFet-A expression is not known. To address this, we treated HepG2 cells with AICAR, an AMPK activator, in the presence of low or high glucose. Treatment of AICAR for 12hr in low glucose or high glucose conditions decreased Fet-A protein synthesis in cell lysates (Fig 2B). In addition, activation of AMPK by AICAR for 12hr dose-dependently decreased secretion of Fet-A and pFetA into the media (Fig 2C and 2D). Secretion of Fet-A from HepG2 cells was confirmed and quantified by ELISA, in which incubation of cells for 12hr with high-glucose increased the secretion of total Fet-A when compared to low glucose, and treatment of AICAR or metformin for 12hr significantly reduced the secretion of total Fet-A (Fig 2E). Fet-A expression was downregulated progressively starting from 2 h (similar time-frame compared to activation of AMPK by AICAR) through 24 h of treatment (S1B Fig). Importantly, gene

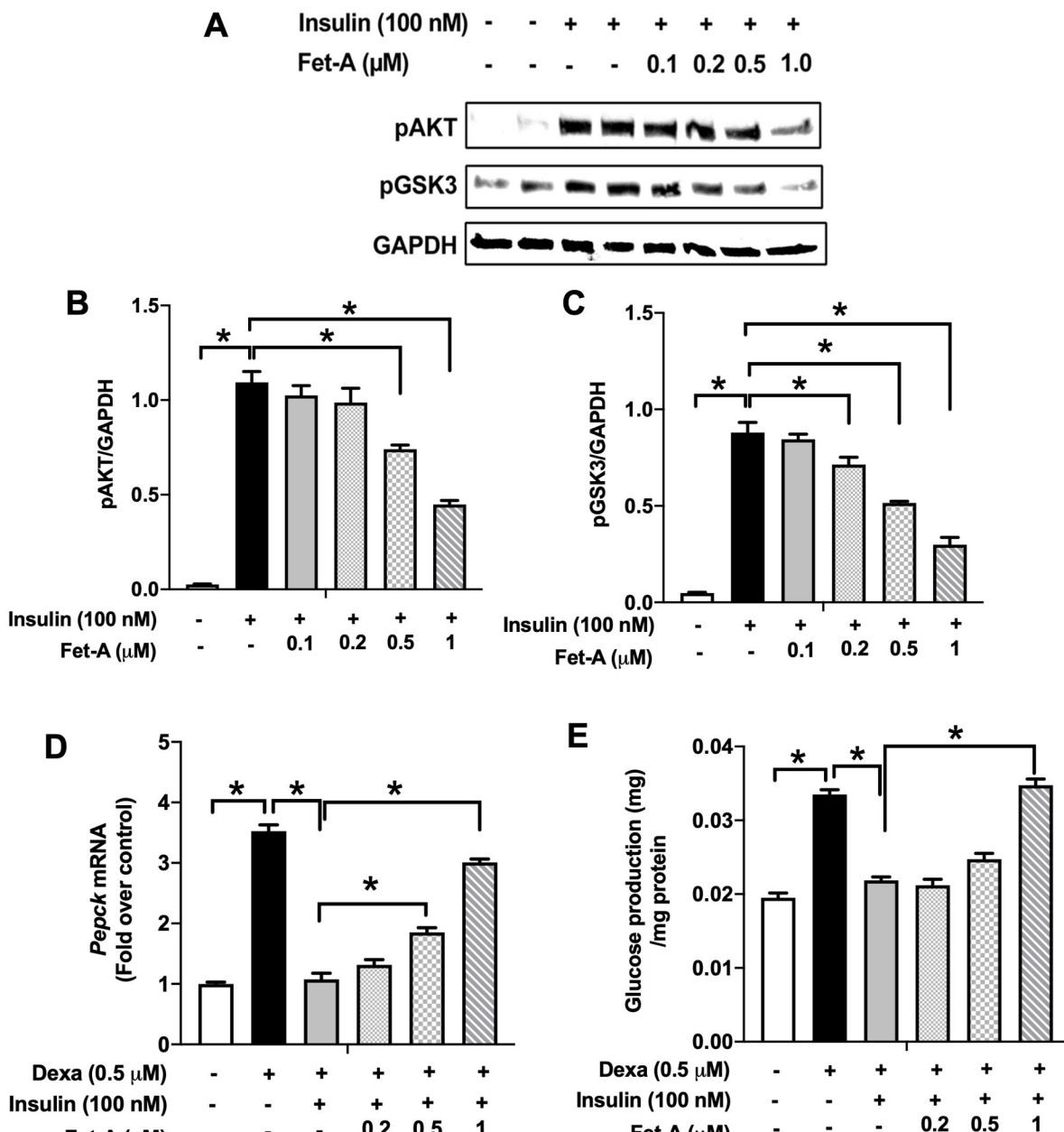

**Fig 1. Effect of fetuin-A (Fet-A) on insulin signaling and insulin mediated suppression of gluconeogenesis and glucose production in HepG2.** [A] HepG2 cells were pre-treated with recombinant Fet-A in the presence or absence of insulin, and cell lysates were subjected to immunoblotting for AKT and GSK3 phosphorylation status (n = 3). [B] Quantified data of the ratio of cellular pAKT/GAPDH immunoblots (n = 3) and C] ratio of cellular pGSK/GAPDH immunoblots (n = 3) are shown. [D] HepG2 cells were serum-starved for 6 h, followed by treatment with dexamethasone (Dexa), insulin, or insulin and Fet-A for 12 h. Real-time gene expression of *Pepck* were analyzed (n = 4). [E] To analyze glucose production, HepG2 cells (n = 4) were treated with 0.5 μM dexamethasone and 0.1 mM 8-CTP-cAMP (Dex/cAMP), various concentrations of Fet-A or 100 nM insulin (Ins) in glucose free DMEM medium (pH 7.4 supplied with 20 mM sodium lactate and 2 mM sodium pyruvate) for 5 h. Glucose production was assayed by measuring glucose concentration in the medium as described previously [33]. Data are shown as Means ± SEM. P values were determined by one-way ANOVA followed by Tukey's multiple comparison tests (A-C). Data are representative of at least three independent experiments performed in replicates. * Indicates p < 0.05.

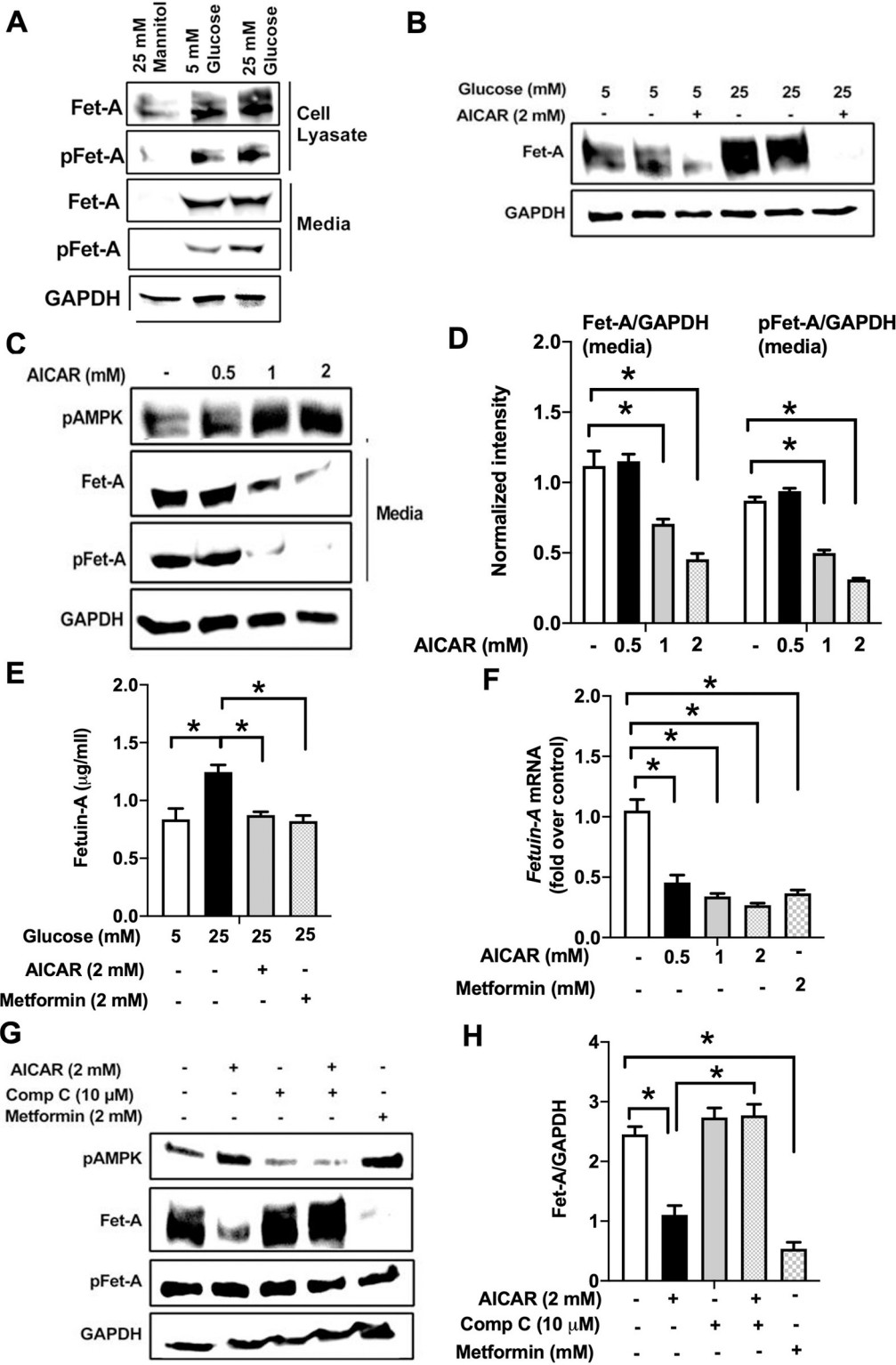

**Fig 2. Activation of AMPK downregulates high glucose-induced Fet-A expression in HepG2 cells.** [A] HepG2 cells were incubated in a media containing either mannitol, low glucose or high glucose for 12 hours, and cell lysates/media were subjected to immunoblotting for Fet-A and pFet-A (n = 3). [B] HepG2 cells were incubated with either low- or high-glucose in the absence or presence of AICAR for 12 h and cell lysates were analyzed by Western blotting. The blots were analyzed with antibodies against Fet-A (n = 4). [C] HepG2 cells were incubated with increasing

concentrations (0.5, 1, 2 mM) of AICAR for 12 h. Cell lysate or media were analyzed by Western blotting for indicated proteins (n = 3) and [D] level of Fet-A and pFet-A in media, as a ratio of GAPDH were expressed. [E] HepG2 cells were incubated in low or high glucose in the absence or presence of AICAR/metformin for 12 hours, and media was used to detect Fet-A by ELISA technique (n = 4). [F] Real-time gene expression analysis was carried out for Fet-A after AICAR and metformin treatment (n = 4). [G] HepG2 cells were incubated with AICAR/metformin in the presence or absence of Compound C, an AMPK inhibitor. Cell lysates were immunoblotted for pAMPK, Fet-A as well as p-Fet-A and [H] Fet-A levels, as a ratio to GAPDH are depicted (n = 4). Data are shown as Means ± SEM. P values were determined by one-way ANOVA followed by Tukey's multiple comparison tests (C-F). Data are representative of at least three independent experiments performed in replicates. * Indicates $p < 0.05$.

expression of Fet-A was also significantly reduced by AICAR or metformin treatment for 12hr (Fig 1F). Furthermore, pretreatment of HeG2 cells with Compound C, an inhibitor of AMPK, reversed the effect of AICAR on Fet-A expression (Fig 2G and 2H). These results indicates that activation of AMPK downregulates Fet-A expression and secretion in HepG2 cell lines.

## AICAR downregulates Fet-A expression through p38 MAPK activation in HepG2 cells

Previously, high glucose was shown to increase the transcriptional activation of Fet-A through ERK 1/2 pathway [25]. To determine the effect of AICAR on high glucose-induced ERK 1/2 expression, we treated HepG2 cells with AICAR in the presence of low- or high-glucose. As shown previously [25], high glucose increased the phosphorylation of ERK 1/2 compared to low glucose (Fig 3A). AICAR treatment further increased ERK 1/2 phosphorylation suggesting that ERK 1/2 pathway may not be involved in AICAR-mediated lowering of Fet-A expression. We observed that AICAR also increased the phosphorylation of other MAP kinases, including JNK and p38 MAPK in a dose-dependent manner (Fig 3B). Pretreatment with p38 MAPK inhibitor SB202190, significantly reversed the effect of AICAR on high glucose-induced expression of Fet-A in cell lysates and its secretion into the media (Fig 3C and 3D). Treatment with Compound C reversed AICAR-induced p38 MAPK activation (Fig 3E and 3F) suggesting that AMPK may regulate Fet-A expression through p38 MAPK. To further confirm the role of p38 MAPK in AMPK-induced reduction in Fet-A expression, we used siRNA to knockdown the expression of p38 MAPK in HepG2 cells. As expected, AICAR treatment decreased Fet-A expression in cells transfected with scrambled siRNA (Fig 3G). However, knockdown (~50%) of p38 MAPK reversed AICAR-mediated inhibition of Fet-A expression (Fig 3G–3I). Interestingly, we observed that AICAR-treatment increased AMPK phosphorylation in both scrambled and p38 MAPK siRNA transfected cells, suggesting that AICAR-induced downregulation of Fet-A is mediated through p38 MAPK activation, downstream of AMPK. To further understand the role of p38 MAPK in the regulation of Fet-A expression, we treated HepG2 cells with p38 MAPK activator, anisomycin. We found that anisomycin treatment, at concentrations previously shown to activate p38 MAPK [37], increased p38 MAPK phosphorylation status and inhibited Fet-A expression (Fig 3J). Since anisomycin is also a protein synthesis inhibitor, we screened other agents including puromycin and cycloheximide, which are known to inhibit protein synthesis without activating p38 MAPK [38]. Unlike anisomycin, the other protein synthesis inhibitors, puromycin and cycloheximide, neither activated p38 MAPK nor inhibited Fet-A expression, suggesting the involvement of p38 MAPK in the regulation of Fet-A (Fig 3J).

## AICAR or anisomycin inhibits Fet-A expression in Hep3B cells and primary rat hepatocytes

To further probe our findings that AMPK and p38 MAPK are involved in the negative regulation of Fet-A, we carried out similar experiments in Hep3B human hepatoma cells and

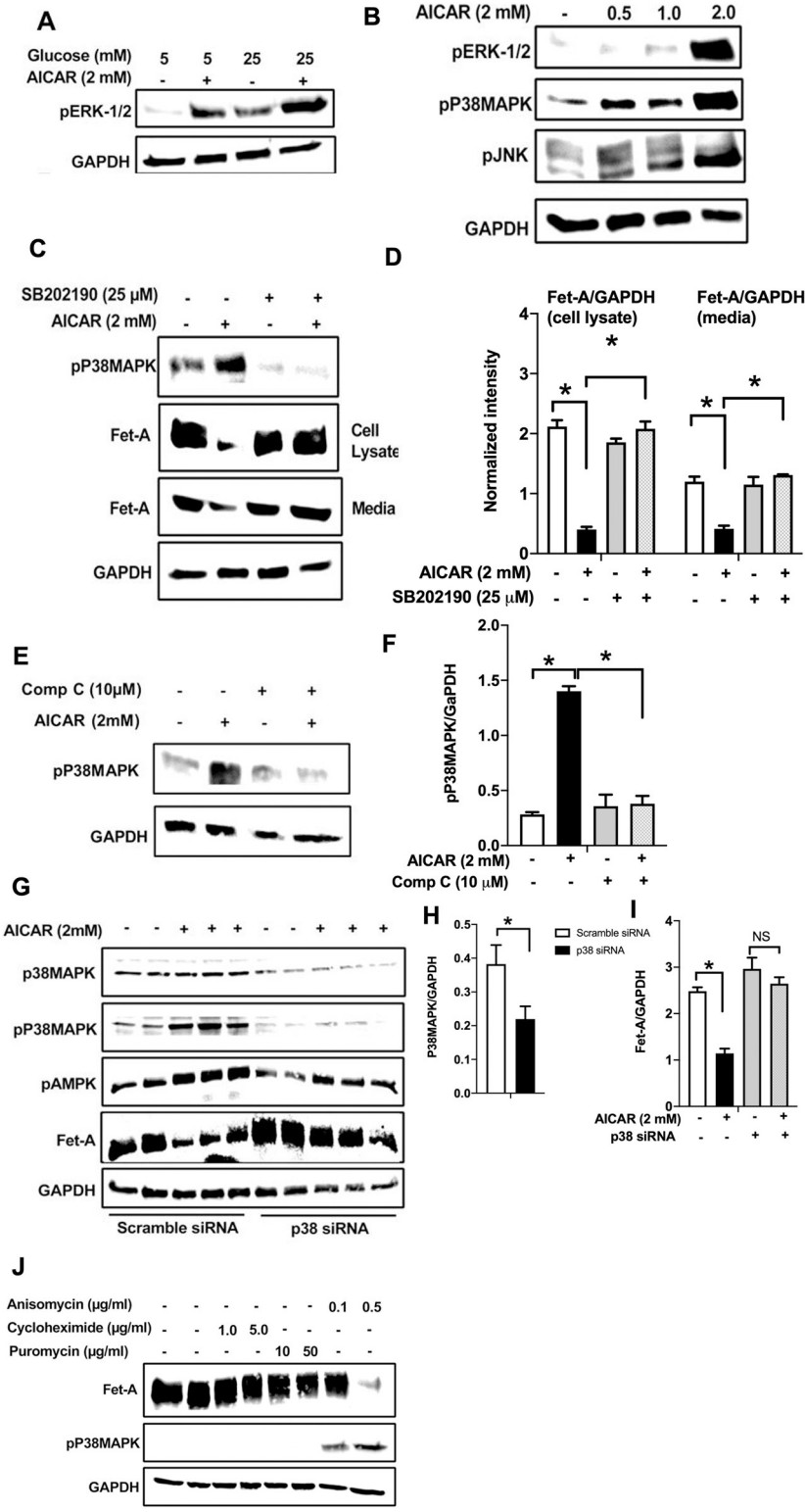

**Fig 3. AMPK activation downregulates Fet-A expression through p38 MAPK.** [A] HepG2 cells were incubated with either low- or high-glucose in the absence or presence of AICAR for 12hr and used for Western blot analysis for ERK1/2 phosphorylation expression (n = 3). [B] HepG2 cells were incubated with different concentration of AICAR for 12 h and cell lysates were analyzed by Western blotting for ERK1/2, p38MAPK and JNK phosphorylation (n = 3). [C] Cells were treated with p38 MAPK inhibitor [SB202190, n = 4] before treatment of AICAR for 12 hr. Cell lysate or media

were analyzed by Western blotting for indicated proteins and [D] Fet-A levels, as a ratio to GAPDH were determined. [E] Cells were treated with AMPK inhibitor [Comp C, n = 4] before treatment of AICAR for 12hr. Cell lysates were analyzed by Western blotting for phosphorylated p38 MAPK (pP38MAPK) and [F] pP38MAPK levels, as a ratio to GAPDH were determined. [G] Knockdown of p38 MAPK was performed using MAPK14 [p38 MAPK] small interfering RNA [siRNA] in HepG2 cells. Following AICAR treatment for 12 h, cell lysates were analyzed by Western blotting for expression of p38 MAPK, phosphorylated p38 MAPK, Fet-A, and pAMPK. [H] Efficiency of p38MAPK siRNA in HepG2 cells were determined by immunoblotting transfected cells for p38MAPK and levels were expressed as a ratio to GAPDH. [I] Effect of AICAR on Fet-A expression in scrambled or p38MAPK siRNA transfected cells were determined by expressing Fet-A levels, as a ratio to GAPDH (n = 4). [J] Effect of protein synthesis inhibitors, cycloheximide and puromycin, were compared with anisomycin, also a protein synthesis inhibitor, for effects on Fet-A and phosphorylated p38 MAPK expression (n = 3). Data are shown as Means ± SEM. P values were determined accordingly by either by unpaired two-tailed test (E) or one-way ANOVA followed by Tukey's multiple comparison tests (C-F). * Indicates $p < 0.05$.

primary rat hepatocytes. Similar to HepG2 cells, high glucose increased the expression of Fet-A and pFet-A in Hep3B cells Fig 4A and 4B and AICAR treatment downregulated it (Fig 4C and 4D) in a dose dependent manner. In addition, direct activation of p38MAPK by anisomycin (Fig 4E and 4F) also decreased Fet-A expression in a dose-dependent manner in Hep3B cells. Furthermore, AICAR treatment led to the downregulation of Fet-A and its secretion into the media, in a concentration dependent manner in primary rat hepatocytes (Fig 4G and 4H). This was consistent with the increase in phosphorylation of AMPK and p38 MAPK (Fig 4G). Further, anisomycin-induced increase in p38 MAPK phosphorylation was associated with a decrease in Fet-A expression (Fig 4I and 4J).

## Proteosomal degradation and C/EBPβ pathway is involved in AICAR induced Fet-A downregulation

To check the involvement of degradation pathway on Fet-A down-regulation, we analyzed the effect of proteasome inhibition on Fet-A expression. MG-132, a proteasome inhibitor, blocked AICAR-induced Fet-A down-regulation at 2 and 4 hr (Fig 5A and 5B). This effect was not observed by MG-132 when AICAR was treated for 12 hr (Fig 5D and 5E), suggesting the involvement of the ubiquitin-proteasome pathway in short-term effect of AICAR (up to 4hr) on Fet-A expression. Next, we tested whether Fet-A was ubiquitinated in cells. We immuno-precipitated Fet-A from cell lysates, and then analyzed with antibody against ubiquitin. As can be seen, the multi-ubiquitination of Fet-A was detected in the immunoprecipitated sample (Fig 5C). These results demonstrated that the proteasomal pathway was involved in AICAR-mediated Fet-A down-regulation.

Next, we considered the role of C/EBPβ on AICAR's inhibitory effect on Fet-A expression, since previous reports have shown several binding sites for C/EBPβ on the promoter region of Fet-A, and that binding of C/EBPβ, augmented Fet-A expression [27]. Here, we observed that 12 hr treatment of AICAR significantly decreased the expression of C/EBPβ in a dose dependent manner in HepG2 cells, (Fig 5F and 5G). This effect was also observed in primary rat hepatocytes (Fig 5H and 5I), suggesting that the effect of AICAR was mediated by a combination of increased degradation of Fet-A by the proteosomal pathway and a decrease in synthesis of Fet-A through C/EBP beta.

## Discussion

In this study, we have explored mechanisms that mediate the hepatic regulation of Fet-A using pharmacological and siRNA-mediated knockdown approaches. While previous studies have shown that high glucose, elevated free fatty acids, dexamethasone, estrogen, and ER stress increase Fet-A expression [14,22–25], this is the first study to examine mechanisms that mediate the downregulation of Fet-A expression (Fig 6).

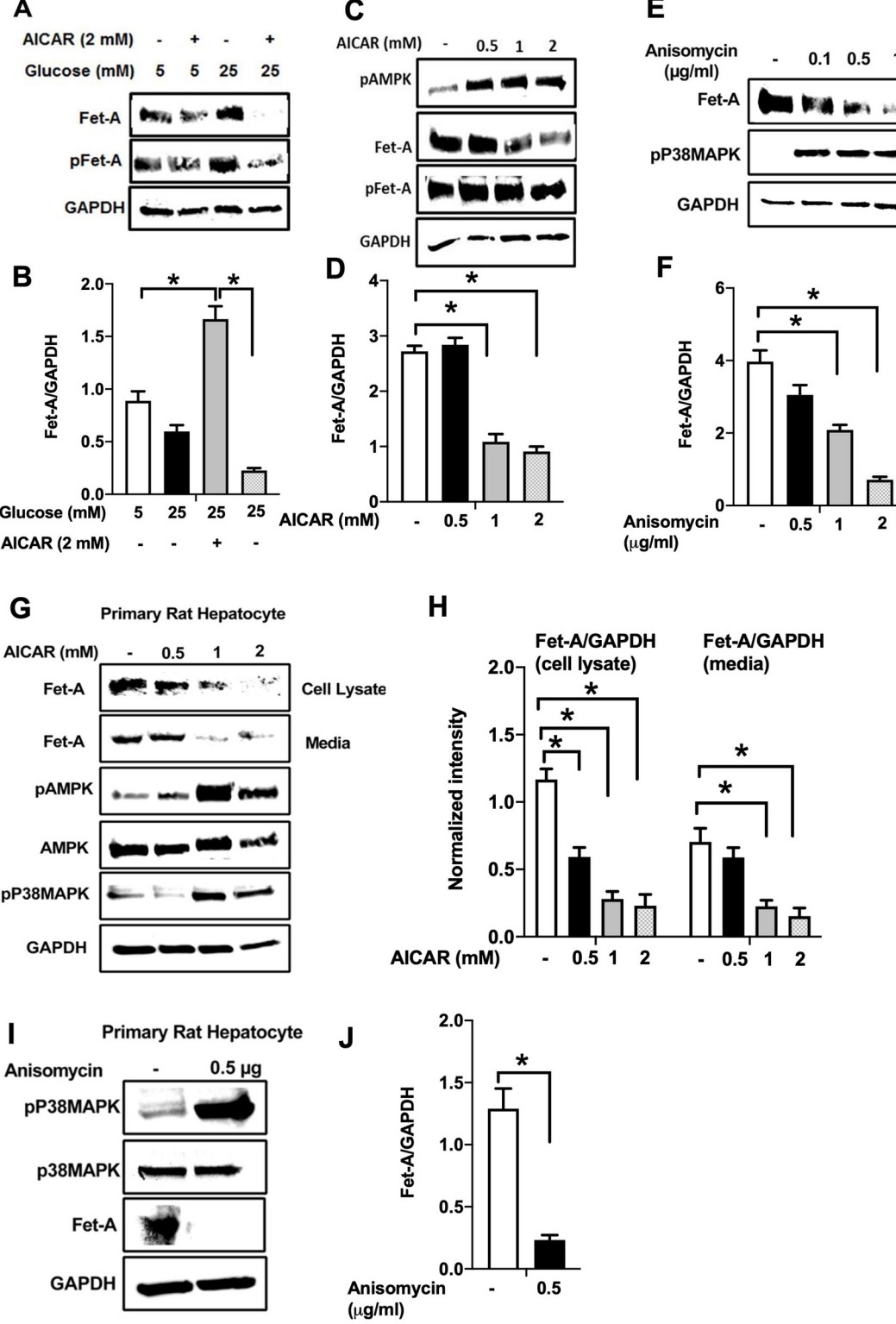

**Fig 4. Effect of AICAR or anisomycin treatment on Fet-A expression in Hep3B cells and primary rat hepatocytes.** [A] Hep3B cells were incubated in low or high glucose media with or without AICAR for 12 hours, and cell lysates were subjected to immunoblotting for Fet-A and phosphorylated Fet-A (n = 4). [B] High glucose-mediated changes in Fet-A level in Hep3B cells was expressed as a ratio to GAPDH. [C] Hep3B cells were incubated with different concentration of AICAR for 12 h and cell lysates were analyzed by Western blotting for indicated proteins (n = 3). [D] Effect of AICAR on Fet-A level in Hep3B cells

was expressed as a ratio of GAPDH. [E] Hep3B cells were treated with various concentrations of anisomycin for 0.5 h to analyze phosphorylated p38 MAPK (pP38MAPK) and Fet-A expression (n = 3). [F] Effect of anisomycin on Fet-A level in Hep3B cells was expressed as a ratio of GAPDH [G] Primary rat hepatocytes were incubated with different concentration of AICAR for 12 h and cell lysate or media were analyzed by Western blotting for indicated proteins (n = 4). [H] Effect of AICAR in primary rat hepatocytes on Fet-A levels were expressed as ratio to GAPDH [I] primary rat hepatocytes were treated with anisomycin for 0.5 h to analyze phosphorylated p38 MAPK (pP38MAPK), Fet-A and p38MAPK expression (n = 4). [J] Effect of anisomycin in primary rat hepatocytes on Fet-A levels were expressed as ratio to GAPDH (n = 4). Data are shown as mean ± SEM. P values were determined accordingly by either by unpaired two-tailed test (E) or one-way ANOVA followed by Tukey's multiple comparison tests (A-D). * Indicates p < 0.05.

In the present study, we have established that AICAR- or metformin-induced activation of AMPK downregulates Fet-A expression in HepG2 and Hep3B human hepatoma cells, and in primary rat hepatocytes. Our studies have also identified p38 MAPK as a critical determinant of hepatic Fet-A expression. Inhibition of p38 MAPK activity using SB202190 or its knock-down with p38 MAPK siRNA blocked AICAR's effect on Fet-A expression. Further, we have shown that direct activation of p38 MAPK with anisomycin inhibited Fet-A expression in HepG2 cells, Hep3B cells and rat primary hepatocytes. Unlike anisomycin, other protein synthesis inhibitors, including cycloheximide and puromycin, had no effect on p38 MAPK and Fet-A expression. A critical question that stems from this study is whether the AICAR/aniso-mycin-mediated downregulation of Fet-A expression is due to a decrease in the synthesis of Fet-A or due to its degradation. Our studies suggest that while this may affect both synthesis and degradation, there may be differences in acute versus long-term regulation. First, our studies show that Fet-A expression was significantly decreased [50–80%] within 2–6 h of AICAR treatment. Second, treatment with protein synthesis inhibitors, puromycin and cycloheximide did not alter Fet-A expression in short-term treatment experiments. Third, inhibition of the proteosomal degradation pathway prevented the short term but not long-term effects of AICAR. This suggests that, acutely, Fet-A undergoes degradation within the cell. However, in a more long-term manner (over 12 hours), AICAR treatment also decreased Fet-A gene expression [synthesis] and its secretion into the media. Since basal levels of Fet-A synthesis in the liver are regulated by C/EBP$\beta$ and NF-1-binding to AHSG promoter [27], we examined long term effect of AICAR on C/EBP$\beta$ expression as indicated by others [39,40]. We observed that long-term AICAR treatment decreases C/EBP$\beta$ expression in HepG2 cells and primary rat hepatocytes, suggesting that C/EBP$\beta$ may be involved in AICAR-induced downregulation of Fet-A synthesis.

Our findings that AICAR-treatment downregulates Fet-A expression is supported by a few indirect studies. Salicylate and adiponectin were shown to inhibit palmitate-induced Fet-A expression through AMPK-NFkB pathway in HepG2 cells [29]. Further, short-term exercise training, which activates AMPK, was shown to significantly decrease plasma Fet-A in subjects with NAFLD [13]. However, Mori et al reported that metformin treatment did not affect Fet-A expression in Fao hepatoma cells and in patients with type 2 diabetes [27,28]. While we have not replicated this study in Fao cells, it is possible that this may be cell-specific effects and/or potential differences in the activation state of AMPK induced by metformin. It is well established that AMPK plays an important role in the regulation of glucose and lipid metabolism and that a low activation state of AMPK has been correlated with metabolic disorders, including insulin resistance and obesity [41–43].

These studies also shed light on the potential differences of secreted Fet-A versus pFet-A. Our studies have shown that high glucose increases pFet-A secretion, but not Fet-A. Recent studies from our laboratory [10,44], implicate a critical role of phosphorylation status of Fet-A in mediating the inhibitory effects of Fet-A on insulin signaling. Our findings raise the possibility that in insulin resistant or hyperglycemic conditions, that pFet-A, the physiologically

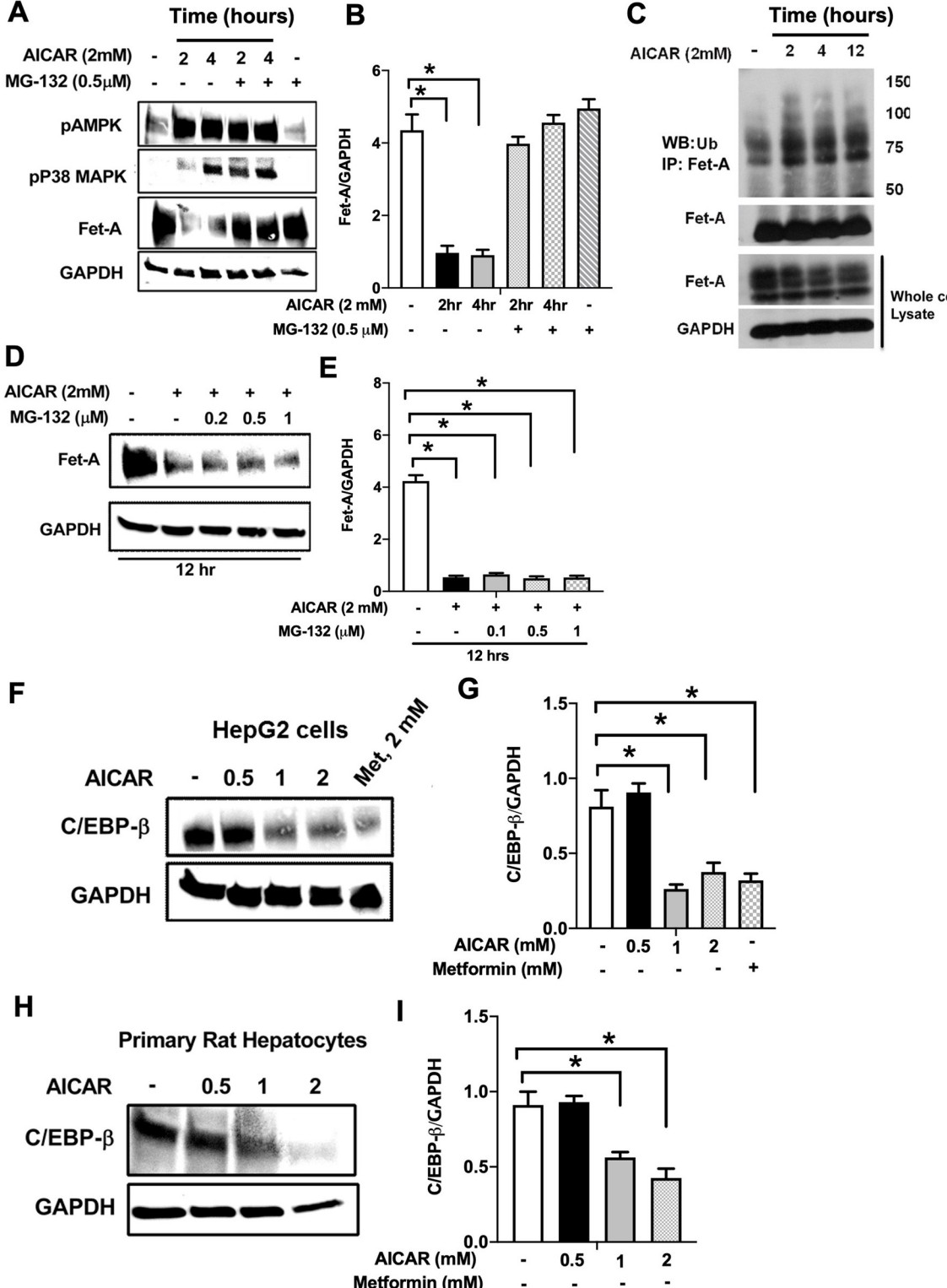

**Fig 5. AMPK regulates Fet-A expression by proteosomal degradation and decreased C/EBP beta expression.** [A] HepG2 cells were treated with AICAR [2mM] at 2 and 4 hr in the presence or absence of MG-132 and used for Western blot analysis of pAMPK, phosphorylated p38MAPK (pP38MAPK), and Fet-A expression (n = 3). [B] Effect of AICAR and MG-132 treatment for 2 and 4 hr on Fet-A levels were expressed as a ratio to GAPDH in HepG2 cells [C] HepG2 cells were treated with AICAR [2mM, 2, 4 and 12 hr] and immunoprecipitated with anti-Fet-A antibody. The immunoprecipitates were analyzed by Western blotting for anti-ubiquitin and Fet-

A antibodies (n = 3). [D] HepG2 cells were treated with AICAR [2mM] in the presence or absence of different concentration of MG-132 for 12 hr and used for Western blot analysis for Fet-A expression (n = 2). [E] Effect of AICAR and MG-132 treatment for 12 hr on Fet-A levels were expressed as a ratio to GAPDH in HepG2 cells. [F] HepG2 cells were incubated with different concentration of AICAR or metformin for 12 hr. Cell lysates were analyzed by Western blotting for C/EBP beta (n = 4) and [G] levels were expressed as a ratio to GAPDH. [H] Primary rat hepatocytes were treated with different concentration of AICAR for 12 hr, lysates were used to immunoblotting for C/EBP beta (n = 4) and [I] levels were expressed as a ratio to GAPDH. Data are shown as Means ± SEM. P values were determined accordingly by one-way ANOVA followed by Tukey's multiple comparison tests (A, C-E). * Indicates p < 0.05.

active form of the inhibitor, may be preferentially secreted by hepatocytes. In this regard, it is of significant interest that our studies show that AICAR treatment decreased the secretion of both Fet-A and pFet-A in HepG2 cells. Additionally, Haukeland et al, demonstrate that metformin treatment for six months decreased serum Fet-A levels in individuals with fatty liver [15]. These results suggest that activation of AMPK can lower elevated Fet-A and potentially pFet-A levels. However, potential mechanisms catalyzing phosphorylation of Fet-A is still unclear.

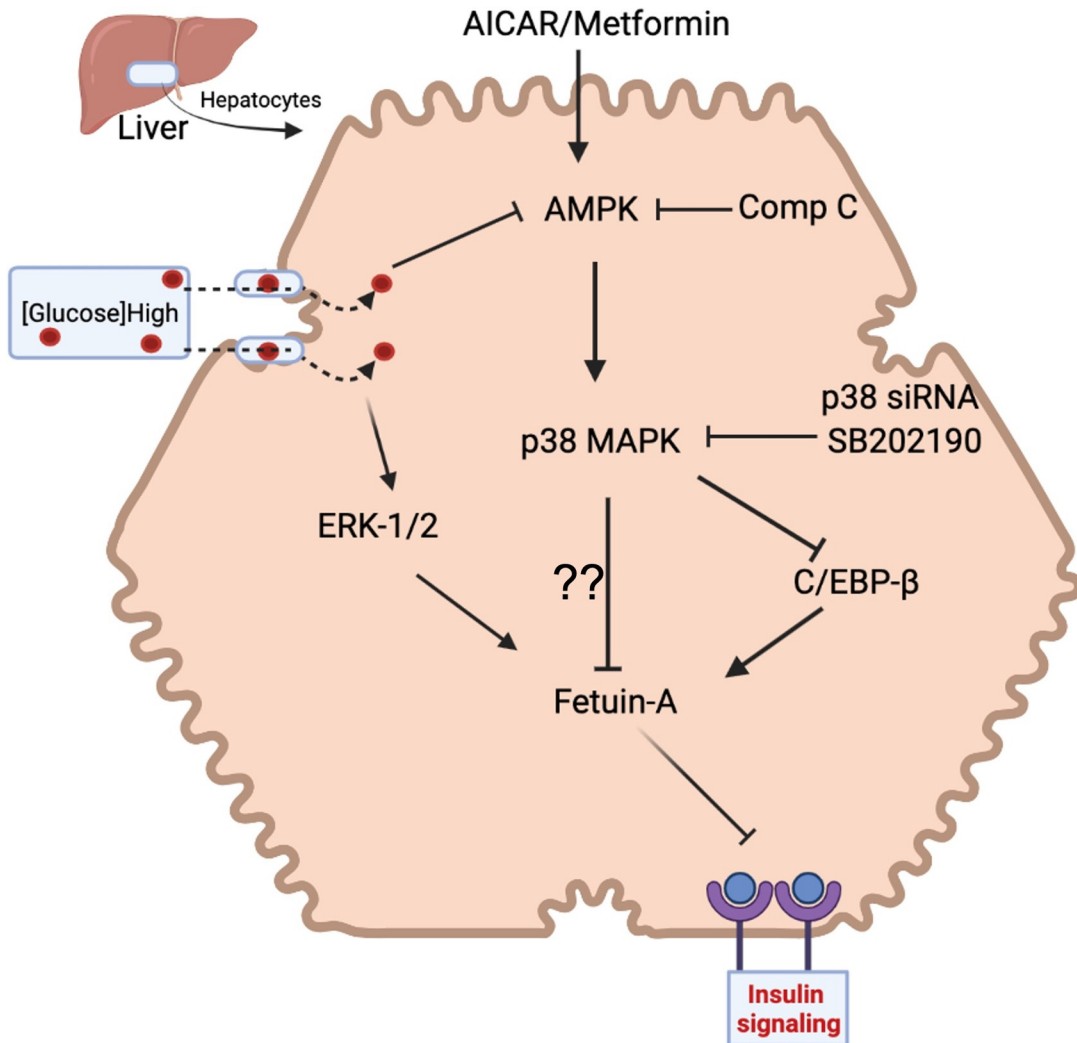

**Fig 6. Working model for AICAR-mediated regulation of hepatic Fet-A expression.** Metformin or AICAR increases the phosphorylation of AMPK, which activates p38 MAPK. As a result, AMPK-p38MAPK signaling suppresses high-glucose induced Fet-A expression either by increasing proteasomal degradation and/or by decreasing hepatic expression of C/EBP beta.

There are several limitations to this study. Signaling proteins upstream of AMPK and downstream of p38 MAPK that mediate the negative regulation of Fet-A remain to be identified. Secondly, while it may be assumed that activation of AMPK and or p38 MAPK pathway may lead to decrease in Fet-A synthesis through C/EBP beta, such mechanisms need to be further characterized. Apart from promoting p38MAPK activation [45–48], AMPK is known to inhibit the mammalian target of rapamycin complex 1 (mTORC1) signaling, which is involved in the regulation of autophagy and protein synthesis [49]. Additionally, recent studies have shown regulation of C/EBPβ-isoform expression by mTORC1 [50]. So, it is important to determine role of mTORC1 in AICAR induced downregulation of hepatic C/EBPβ-isoform and characterize their relationship with Fet-A expression. Future studies will include characterization of the proposed mechanisms shown above. Additionally, these will require validation in a relevant animal model.

In conclusion, our studies provide strong biochemical evidence for a novel AMPK→ p38 MAPK pathway in the regulation of hepatic Fet-A expression in both low- and high glucose conditions. Our studies show that the effect of AICAR is mediated by a combination of an increase in proteosomal degradation of Fet-A and a decrease in synthesis of Fet-A through C/EBP beta. This is the first characterization of the mechanisms that downregulate Fet-A, a protein shown to be associated with obesity, liver fat, insulin resistance, metabolic syndrome, and incident diabetes [51–54].

## Supporting information

**S1 Fig. High glucose impairs insulin signaling through GSK3 and time-course of Fet-A downregulation by AICAR treatment in HepG2 cells.** HepG2 cells were incubated in low (5 mM) or high glucose (25 mM) media for 12 h followed by insulin treatment for 15 min and cell lysate were subjected to immunoblotting for GSK3 phosphorylation status. [B] The effect of AICAR (2 mM) on HepG2 cells under high glucose condition at different periods was observed by Western blot analysis.
(TIF)

**S1 Raw images.**
(PDF)

## Author Contributions

**Conceptualization:** Jeganathan Ramesh Babu, Suresh T. Mathews.

**Data curation:** Suresh T. Mathews.

**Formal analysis:** Vishal Kothari.

**Funding acquisition:** Suresh T. Mathews.

**Methodology:** Vishal Kothari.

**Project administration:** Suresh T. Mathews.

**Resources:** Jeganathan Ramesh Babu.

**Supervision:** Suresh T. Mathews.

**Writing – original draft:** Vishal Kothari.

**Writing – review & editing:** Jeganathan Ramesh Babu, Suresh T. Mathews.

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
