## [Decision Letter · Decision Letter 0]

13 Jan 2022

PONE-D-21-38362AMP activated kinase negatively regulates hepatic Fetuin-A via p38 MAPK-C/EBPβ/E3 Ubiquitin Ligase Signaling Pathw ayPLOS ONE

Dear Dr. Mathews,

Thank you for submitting your manuscript to PLOS ONE. After careful consideration, we feel that it has merit but does not fully meet PLOS ONE’s publication criteria as it currently stands. Therefore, we invite you to submit a revised version of the manuscript that addresses the points raised during the review process. Your manuscript has been evaluated by two referees and their comments are available below. Please be sure to clarify the statistical analyses. This includes to perform the statistical analyses for all immunoblots and indicate in the figures the results of these analyses. Additionally, a detailed explanation of the symbols used to indicate statistical significance is required in figure caption. For complex blots, it is recommended to include a bar graph, placed next to the blot, that depicts the mean and variance, and captures the statistical analyses. There is some concern that the blots are overexposed. The authors are invited to submit blots with pixelated background. Additionally, be sure to validate all concentration-dependent effects by performing the appropriate statistical analysis and report the results of the analyses in the figures and their caption. The reviewers request significant improvements to the reporting of methodological aspects of the study, especially with the transfection experiments, reagent sourcing, treatment exposure time, and sample size. To address these comments, please be sure to include reagent catalogue numbers, include exposure times wherever it would help the readers, and insert the sample size in figure captions.

We look forward to receiving your revised manuscript.

Kind regards,

Regis Moreau, Ph.D.

Academic Editor

PLOS ONE

Journal Requirements:

 [This study was partly supported by the United States Department of Agriculture, National Institute of Food and Agriculture grant ALA043-1-08033 (to S.T.M) No additional funding was received for this study. The funders had no role in study design, data collection and analysis, decision to publish, or preparation of the manuscript.]

[This work was supported in whole or in part, by the United States Department of Agriculture, National Institute of Food and Agriculture grant ALA043-1-08033 (to S.T.M)]

[This study was partly supported by the United States Department of Agriculture, National Institute of Food and Agriculture grant ALA043-1-08033 (to S.T.M) No additional funding was received for this study. The funders had no role in study design, data collection and analysis, decision to publish, or preparation of the manuscript.]

5. Please amend either the title on the online submission form (via Edit Submission) or the title in the manuscript so that they are identical.

Reviewers' comments:

Reviewer's Responses to Questions

**Comments to the Author**

1. Is the manuscript technically sound, and do the data support the conclusions?

Reviewer #1: Yes

Reviewer #2: Partly

2. Has the statistical analysis been performed appropriately and rigorously? 

Reviewer #1: No

Reviewer #2: No

3. Have the authors made all data underlying the findings in their manuscript fully available?

Reviewer #1: Yes

Reviewer #2: No

4. Is the manuscript presented in an intelligible fashion and written in standard English?

Reviewer #1: Yes

Reviewer #2: Yes

5. Review Comments to the Author

Reviewer #1: In the presented manuscript, the authors investigated whether activation or inhibition of AMPK may influence the Fetuin-A hepatokine expression and activity in liver cells. They found that AMPK negatively regulates via p38 MAPK-C/EBPβ/E3 Ubiquitin Ligase Signalling Pathway. This study if of high interest, and brings new insights in the understanding of Fetuin-A associated pathways that lead to insulin resistance in peripheral tissues including liver. The experiment is well designed and includes appropriate control groups, which is very important. Before this manuscript could be accepted for publication, authors need to address the following comments:

1. In the transfection method, more details have to be provided; which siRNA has been used? How transfection efficiency has been evaluated?

2. How authors explain the absence of protein band for Fetuin-A protein in the control group while liver cells are known to secrete the protein even under normal physiological conditions.

3. Figure 1: [A] HepG2 cells were pre-treated with recombinant Fet-A in the presence or absence of insulin and cell lysates were subjected to immunoblotting for AKT and GSK3 phosphorylation status. Why authors didn’t evaluate insulin signalling pathway after High-glucose cells challenging

4. Authors need to include INSR protein (Insulin receptor) in their panel, as Fetuin-A is known to directly act on the phosphorylation status of INSR.

5. Authors need to normalize the protein expression results to GAPDH and provide Relative gene expression data that will bring more reliable conclusions. Immunoblots are insufficient for such conclusions. After Protein relative expression calculation, statistical analysis needs to be done as well.

6. The density of some bands is very high which could strongly influence the results and make normalization tedious. Can you provide better blots?

7. Figure 1.A, can authors explain why there are two groups with Insulin + and Fetuin-A – and Insulin – Fetuin-A -? What is the difference in the two groups? This is confusing.

8. Figure 1.F, which method has been used to calculate Fet-A gene expression?

9. In some immunoblots, the GAPDH density is not the same from one group to another one, are the authors sure they loaded in the electrophoresis gel the exact same number of proteins?

10. Gene expression analysis of some effectors such as p38MAPK and ERK would bring more value of the present experiment.

Reviewer #2: The opinion of bioethics committee was not mentioned, please provide it if exists. The time of your research study was no indicated. How long they last? What is the sample? The preliminary characteristics of the variables analyzed are missing.

The statistical tests were indicated but your description needs to be complemented with an information wherher you use the unpaired Student's t-test or one-wayAnalysis of Variance (ANOVA), in order to avoid the repetition (for ex. in verse 203/204). After analysis with the AVOVA test, which provides differences in parameter level, it should be pointed out in which group the parameter achieves the highest and lowest value (post-hoc test, p-value).

In verse 204 indicates P < 0.05 should be replaced by indicates p < 0.05.

I suggest a summary table to present the Mean ± Standard Error of the Mean (SEM) and p-value for the analysed parameters. The graphic in its current shape is unclear.

In Results, a reference to indicate p-value is lacking (appears only in verse 204) and the analysis also is not summarised which has a negative impact on the whole.

As indicated in verse 253 /254 „This was correlated with anincrease in phosphorylation of AMPK and p38 MAPK (Fig 3D), but lack of information about correlation analysys in statistical method.

6. PLOS authors have the option to publish the peer review history of their article (what does this mean?). If published, this will include your full peer review and any attached files.

Reviewer #1: No

Reviewer #2: No

---

## [Author Response · Author response to Decision Letter 0]

9 Feb 2022

We thank the reviewers for their constructive comments and suggestions, which have helped us to markedly strengthen and improve the manuscript. We have carefully addressed all comments of the reviewers. We hope this revised manuscript will merit a favorable response by the reviewers and the Editor. We are submitting both a clean copy of the revised manuscript as well as one with Track changes, as required.

A point-by-point response to the reviewers’ comments are shown below.

Reviewer #1’s comments: 

In the presented manuscript, the authors investigated whether activation or inhibition of AMPK may influence the Fetuin-A hepatokine expression and activity in liver cells. They found that AMPK negatively regulates via p38 MAPK-C/EBPβ/E3 Ubiquitin Ligase Signalling Pathway. This study if of high interest and brings new insights in the understanding of Fetuin-A associated pathways that lead to insulin resistance in peripheral tissues including liver. The experiment is well designed and includes appropriate control groups, which is very important. Before this manuscript could be accepted for publication, authors need to address the following comments:

1. In the transfection method, more details have to be provided; which siRNA has been used? How transfection efficiency has been evaluated?

Response: Additional information regarding p38MAPK siRNA transfection experiments have been included on page 8. Transfection efficiency of the siRNA has been evaluated by immunoblotting p38MAPK expression. This is now included as Fig. 3G,H in the revised manuscript, which shows that transfection of siRNA reduced the expression of p38MAPK by ~50%.

2. How authors explain the absence of protein band for Fetuin-A protein in the control group while liver cells are known to secrete the protein even under normal physiological conditions. 

Response: Yes, in normal physiological condition, fetuin-A is known to be secreted by the liver. Consistent with it, we observed that HepG2 cells cultured in either 5mM glucose or 25mM glucose secrete fetuin-A and its phosphorylated form abundantly (Fig 2A and Fig 2C). However, cells cultured only in mannitol (not physiological) in the absence of glucose does not secrete Fet-A into the media (Fig 2A), as shown in the manuscript

3. Figure 1: [A] HepG2 cells were pre-treated with recombinant Fet-A in the presence or absence of insulin and cell lysates were subjected to immunoblotting for AKT and GSK3 phosphorylation status. Why authors didn’t evaluate insulin signaling pathway after High-glucose cells challenging. 

• Response: Thanks for suggesting this experiment. We have now evaluated the effect of high glucose on insulin signaling pathway in HepG2 cells. As shown in Supplementary data, Suppl. Figure 1A, high-glucose (25mM) impaired phosphorylation of AKT and GSK3 as compared to low glucose (5 mM). This is consistent with higher expression of Fetuin-A in high-glucose treated cells. Our observation of impaired insulin signaling in high glucose treated HepG2 cells is consistent with previously published papers (PMID: 31634780, 30355754).

4. Authors need to include INSR protein (Insulin receptor) in their panel, as Fetuin-A is known to directly act on the phosphorylation status of INSR.

Response: Yes, this was previously discovered by us (Suresh Mathews) and later confirmed by other researchers that, fetuin-A can directly inhibit insulin receptor phosphorylation in vitro, in intact cells, and in vivo (PMID: 11026561, PMID: 31084489, PMID: 17011519, PMID: 12145157). To extend this observation on downstream insulin signaling here we next analyzed the effect of fetuin-A on insulin’s suppressive action of gluconeogenesis and glucose production. As expected, dexamethasone induced a significant increase in gene expression of phosphoenol pyruvate carboxykinase (Pepck) and Insulin treatment significantly inhibited dexamethasone-induced expression of Pepck by over 95%. While, fetuin-A dose-dependently suppressed the insulin effect on gene expression of Pepck (Fig 1D). Similarly, fetuin-A significantly impaired the insulin effect of dexamethasone-induced glucose production (Fig 1E). These data indicates that fetuin-A can impair the insulin signaling at gene expression, protein as well as functional level. This data is shown below and is included in the revised manuscript (as Fig.1D and 1E). 

5. Authors need to normalize the protein expression results to GAPDH and provide Relative gene expression data that will bring more reliable conclusions. Immunoblots are insufficient for such conclusions. After Protein relative expression calculation, statistical analysis needs to be done as well.

Response: Thank you for your suggestion. For all important experiments, we have now included normalization data of relative protein expression of fetuin-A to GAPDH. Additionally, the number of experiments done for each assay and the statistical analyses (p values) are now shown in the revised manuscript. 

6. The density of some bands is very high which could strongly influence the results and make normalization tedious. Can you provide better blots? 

Response: Fetuin-A exist in different glycosylated and phosphorylated forms. Although majority of the fetuin-A exists as monomer, small fractions are known to have the expected molecular weight for the dimer. Because of these varied glycosylation and phosphorylation status, we observed that it is difficult to get the single clean low-density band for fetuin-A. We have been working in this field for several years and images are consistent with a typical broad band around 60 kDa. 

7. Figure 1.A, can authors explain why there are two groups with Insulin + and Fetuin-A – and Insulin – Fetuin-A -? What is the difference in the two groups? This is confusing. 

Response: To understand the baseline expression AKT and GSK3 before and after insulin treatment, we ran two different samples in the same western blot. Sorry about confusion and now relative expression results to GAPDH is now shown. We hope this clear any confusion. 

8. Figure 1.F, which method has been used to calculate Fet-A gene expression?

Response: Thank you for pointing out this as we missed to mention the calculation method. Now, we described in method section, page 9 (line 177) that gene expression of Fet-A has been determined by real time PCR and relative changes in mRNA expression were calculated using the ∆∆Ct method, normalized to beta-actin.

9. In some immunoblots, the GAPDH density is not the same from one group to another one, are the authors sure they loaded in the electrophoresis gel the exact same number of proteins? 

Response: Yes, in all experiment we loaded the same protein amount across all group. Although most experiment shows similar GAPDH density, few immunoblots have minor differences in GAPDH density. We have carried out normalization of fetuin-A with corresponding GAPDH density. The overall data (outcome and interpretation) has not changed. 

10. Gene expression analysis of some effectors such as p38MAPK and ERK would bring more value of the present experiment.

Response: We understand that more analysis will add more value to the experiment and would help to further understand the regulation of fetuin-A in hepatocytes. However, activity of kinases such as p38MAPK and ERK known to be regulated by its phosphorylation status. Thus, measurement of phosphorylation status of effectors such p38 MAPK and ERK (at the protein level) will be more helpful than gene expression. 

Reviewer #2: 

• The opinion of bioethics committee was not mentioned, please provide it if exists.

Response: No human embryonic stem cells were used in this study and therefore no specific registration number exists. Only primary rat hepatocytes were used in this study, which were commercially obtained (Triangle Research Labs). Since no animals were used directly in the study, an IACUC approval from the university was not required. 

• The time of your research study was no indicated. How long they last? What is the sample? The preliminary characteristics of the variables analyzed are missing.

Response: The revised manuscript now shows the treatment time; please see Methods section on page 7 (line 135, 139), and in all figure legends, to better explain treatment time. Cells were treated for 12hr with AICAR or metformin under high glucose condition, unless otherwise indicated. 

To further demonstrate and understand the duration of the effect of AICAR on Fet-A expression, we performed a time-course study. As described in Supplementary data, Suppl. Figure 1B, Fet-A expression was downregulated in HepG2 cells progressively starting from 2 h (similar timeframe compared to activation of AMPK by AICAR) through 24 h of treatment. pFet-A concentrations were not affected during this treatment period, except for 24 h.

• The statistical tests were indicated but your description needs to be complemented with an information whether you use the unpaired Student's t-test or one-way Analysis of Variance (ANOVA), in order to avoid the repetition (for ex. in verse 203/204). After analysis with the AVOVA test, which provides differences in parameter level, it should be pointed out in which group the parameter achieves the highest and lowest value (post-hoc test, p-value). I suggest a summary table to present the Mean ± Standard Error of the Mean (SEM) and p-value for the analysed parameters. The graphic in its current shape is unclear

Response: We thank the reviewer for pointing this out. We have now analyzed relative gene expression of Fetuin-A to GAPDH and determine the statistical analysis either by unpaired Student's t-test or one-way Analysis of Variance (ANOVA) with posthoc analysis. In addition, we indicated in figures which group achieve significant difference compared to other treatment group (p < 0.05). In every figure legend, we have now indicated ‘n’ number of times experiment was done, and which test was applied for specific experiment to make it clearer. 

• In verse 204 indicates P < 0.05 should be replaced by indicates p < 0.05. 

Response: This is now corrected

• In Results, a reference to indicate p-value is lacking (appears only in verse 204) and the analysis also is not summarized which has a negative impact on the whole. As indicated in verse 253 /254 „This was correlated with an increase in phosphorylation of AMPK and p38 MAPK (Fig 3D), but lack of information about correlation analysis in statistical method.

Response: Statistical analyses of all major experiments are now shown in figures, figure legends, and in the results section. Further, we are sorry about confusion in verse 253 /254 where we accidentally used the word “correlated”. The text is now changed to read “This was consistent with the increase in phosphorylation of AMPK and p38 MAPK” (lines 329-330 in revised document).

---

## [Decision Letter · Decision Letter 1]

22 Mar 2022

AMP activated kinase negatively regulates hepatic Fetuin-A via p38 MAPK-C/EBPβ/E3 Ubiquitin Ligase Signaling Pathway

PONE-D-21-38362R1

Dear Dr. Mathews,

We’re pleased to inform you that your manuscript has been judged scientifically suitable for publication and will be formally accepted for publication once it meets all outstanding technical requirements.

Kind regards,

Regis Moreau, Ph.D.

Academic Editor

PLOS ONE

Additional Editor Comments (optional):

Reviewers' comments:

Reviewer's Responses to Questions

**Comments to the Author**

1. If the authors have adequately addressed your comments raised in a previous round of review and you feel that this manuscript is now acceptable for publication, you may indicate that here to bypass the “Comments to the Author” section, enter your conflict of interest statement in the “Confidential to Editor” section, and submit your "Accept" recommendation.

Reviewer #1: All comments have been addressed

2. Is the manuscript technically sound, and do the data support the conclusions?

Reviewer #1: Yes

3. Has the statistical analysis been performed appropriately and rigorously? 

Reviewer #1: Yes

4. Have the authors made all data underlying the findings in their manuscript fully available?

Reviewer #1: Yes

5. Is the manuscript presented in an intelligible fashion and written in standard English?

Reviewer #1: Yes

6. Review Comments to the Author

Reviewer #1: The authors properly addressed all my comments, therefore, I recommend this manuscript for publication

7. PLOS authors have the option to publish the peer review history of their article (what does this mean?). If published, this will include your full peer review and any attached files.

Reviewer #1: No

---

## [Editor Report · Acceptance letter]

29 Apr 2022

PONE-D-21-38362R1 

AMP activated kinase negatively regulates hepatic Fetuin-A via p38 MAPK-C/EBPβ/E3 Ubiquitin Ligase Signaling Pathway 

Dear Dr. Mathews:

I'm pleased to inform you that your manuscript has been deemed suitable for publication in PLOS ONE. Congratulations! Your manuscript is now with our production department. 

Kind regards, 

on behalf of

Dr. Regis Moreau 

Academic Editor

PLOS ONE